



# Effect of deforestation and subsequent land-use management on soil carbon stocks in the South American Chaco

Natalia Andrea Osinaga[1]; Carina Rosa Álvarez [2], Miguel Angel Taboada[1,2,3]

[1]CONICET. National Council of Scientific and Technical Research.
[2]University of Buenos Aires, School of Agronomy, Soil Fertility and Fertilizer. Av. San Martín 4453, Ciudad Autónoma de Buenos Aires, Argentina. 1417.
[3]Soil Institute. CIRN, INTA.

*Correspondence to*: Carina Rosa Alvarez (alvarezc@agro.uba.ar)

**Abstract.** The sub-humid Chaco region of Argentina, originally covered by dry sclerophyll forest, has been subjected to clearing since the end of the '70 and replacement of the forest by no till farming. Land use changes produced a decrease in aboveground carbon stored in forests, but little is known about the impact on soil organic C stocks. The aim of this study was

to evaluate soil C stocks and C fractions up to 1 m depth in soils under different land use: < 10 yr continuous cropping; > 20 yr continuous cropping, warm season grass pasture and native forest in 32 sites distributed over the Chaco region. The organic C stock content up to 1 m depth expressed as equivalent mass varied as follows: forest (119.3 Mg ha$^{-1}$) > pasture (87.9 Mg ha$^{-1}$) > continuous cropping (71.9 and 77.3 Mg ha$^{-1}$), with no impact of the number of years under cropping. The most sensitive organic carbon fraction was the coarse particle fraction (2000 μm -212 μm) at 0-5 cm and 5-20 cm depth

layers. Resistant carbon (<53 μm) was the main organic matter fraction in all sample categories except in the forest. Organic C stock, its quality and distribution in the profile were sensitive to land use change. The conversion of the Chaco forest to crops was associated to a decrease of Organic C stock up to the meter depth and with the decrease of the labile fraction. The incorporation of pastures of warm-season grasses was able to mitigate the decrease of C stocks caused by cropping and so could be considered a sustainable management practice. As soil organic carbon losses were not restricted to the first few cm

of the soil, the development of models that would allow the estimation of soil organic carbon changes in depth would be useful to evaluate with greater precision the impact of land use change on carbon stocks.

Keywords: Carbon sequestration, particulate carbon, land-use change

## 1. Introduction

As one of the components of global change, land use change has a great impact on terrestrial ecosystems, altering their structure and function (Walker and Steffen, 1999). The most important land use change is due to agriculturization (Houghton, 1999), a process that involves replacement of natural ecosystems, such as forests, as world food demand increases (Volante et al., 2012).



In Argentina, since the late 1970s, there has been an advance of the agricultural frontier across the Chaco-region native forests due to conversion for production of annual crops (Gasparri et al., 2009). Thus it became one of the ten countries with the greatest forest loss in the world (FAO, 2015). The Eastern Subhumid Chaco is a large forest area that since 1997 has suffered a notable increase in cleared area (Albanesi et al., 2003; Grau et al., 2005; Volante et al., 2009). The average

deforestation rate is among the highest in the world and in the country, mainly in the East of the province of Santiago del Estero, where Mollisols are the predominant soil type (Volante et al., 2009).

Deforestation together with inadequate subsequent management produces acceleration of erosive processes, reduction of organic matter input, decrease of soil aggregate stability (Cerdà, 2000; Cerdà et al., 2009; García Orenes et al., 2010), changes in microclimate, biodiversity loss, affects water basin functions and contributes to global climate change. These

effects have been studied mostly in tropical and temperate forests but have been poorly evaluated in South America subtropical forests (Baccini et al., 2012; Harris et al., 2012; Hansen et al., 2013).

In the Subhumid Chaco, the intensity and seasonality of rainfall, the gently undulating landscape, the fragility of the environment and the subtropical climate predispose the soil to substantial physical degradation (Albanesi et al., 2003). No-tillage was introduced in Argentina, including in the Chaco region, in the mid-nineties. It was adopted due to its lower

production costs, the possibility it offered of incorporating areas with greater limitations to crop yield (Satorre, 2005; Derpsch et al., 2010), to savings in operating time and to lack of soil disturbance that reduces soil erosion, recovers soil aggregate stability, conserves water and increases carbon sequestration (Díaz Zorita et al., 2002). Despite its many advantages, no-tillage can negatively impact some physical properties of the surface soil (bulk density, penetration resistance), as mechanical formation of macropores is reduced and there is a tendency to form laminar and massive

structures (Strudley et al., 2008; Álvarez et al., 2009; 2012). All these effects are increased by the transit of heavy machinery that produces soil compaction of the first 40 cm of soil, especially when the soil is wet (Botta et al., 2004).

In the western part of the region, livestock production became important, replacing native forests by megathermic pastures. This activity has negative effects on soil physical properties and produces reduction of soil organic carbon levels due to forest clearance and animal transit (Caruso et al., 2012). However, it could have a smaller negative impact than continuous

agriculture on carbon sequestration and on soil physical properties as animal trampling effects extend to a lesser depth and live roots are present in the soil all year long.

The objective of the present study was to determine carbon content and soil physical quality of Subhumid Chaco soils under different land uses: agriculture (less than 10 years and more than 20 years under cropping), pastures and natural forests.

**2. Materials and Methods**

The region of the sub-humid Chaco is part of the Great American Chaco and occupies the southern fringe of the eastern part of the semi-arid Chaco (Vargas Gil, 1988, Figure 1). In Argentina it covers an area of 45,199.33 km[2]. Annual rainfall ranges from 700 mm in the West to 1000 mm on the limit with the humid Chaco (East), it has a monsoon regime, with periods of marked water deficit during the winter months and the beginning of spring. Average annual temperature is 21 ° C. The most



representative soils are Haplustolls and Argiustolls (Vargas Gil, 1988). Crop production is mainly summer crops (soybean, corn, sorghum and cotton) sown in December and January, with winter months generally as fallow periods, in order to store soil water for the summer crop. In the west of the region, livestock production on megathermic pastures predominates. The natural vegetation is a xerophitic forest with dominance of various species of *Schinopsis*, *Prosopis nigra*, *Zizyphus mistol*

5    and shrubs of the genus *Acacia*.

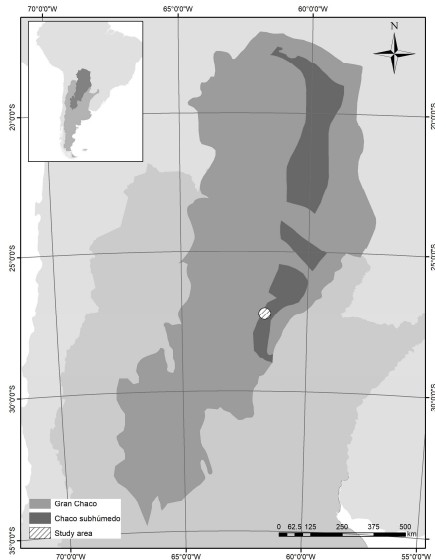

Figure 1: Location of the Gran Chaco, the Subhumid Chaco and the study area.

A total of 32 sites were selected in an area of 320,000 ha to the E of Santiago del Estero province (Figure 1), which were

10   representative of the most common forms of land use of this region: native forest (reference), continuous cropping (rotation of soybean-soybean-corn under no-till) during different periods  (6-9 years and > 20 years) and more than 10-year old pastures (pasture, Gatton Panic, *Panicum maximum*) on the most representative soils (typical Haplustolls and Argiustolls) with silty and clayey texture (Table 1). From each category, 8 sites (n = 8) located in different farms were sampled.

In each situation, composite samples were taken up to 1m depth from the 0-5 cm layer, the 5-20 cm layer, and then every 20

15   cm. Soil organic carbon (SOC) was determined by wet combustion using the Walkley-Black method (Nelson & Sommers, 1996) and coarse particulate organic carbon (2000 μm - 212 μm, CPC), fine particulate organic carbon (212 μm - 53 μm, FPC) and resistant organic carbon (<53μm, RC) were determined (Cambardela & Elliot, 1992). Bulk density (BD) was determined by the cylinder method (Burke et al., 1986) using 100 cm$^3$ cylinders. Carbon content mass per unit area was estimated using sample BD values.



**Table 1: Main characteristics of the soils of the study region.**

| | Soil Order | Area (%) | Horizon | Clay (g kg$^{-1}$) | Silt (g kg$^{-1}$) | Sand (g kg$^{-1}$) | pH |
|---|---|---|---|---|---|---|---|
| Typic Haplustoll | Mollisol | 80 | A | 20 | 43 | 37 | 6.6 |
| | | | AC | 35 | 47 | 18 | 6.7 |
| Typic Argiustoll | Mollisol | 20 | A | 25 | 47 | 28 | 6.8 |
| | | | Bt | 41 | 40 | 19 | 6.6 |

Additionally, four soil sub-samples from the 0-20cm layer were taken in each situation to determine the structural stability

according to the methodology described by Le Bissonnais (1996). The aggregate mean weight diameter (MWD) was used as an index of structural stability. Penetration resistance was determined every 5 cm up to 40 cm depth with a 30º-conical tip dynamic penetrometer (Burke et al., 1986), taking 4 determinations per plot. At the same time soil water content was determined at two depths (0-20 and 20-40 cm), as penetration resistance varies with it. Data were analyzed using analysis of variance (ANOVA) after checking data normality (Shapiro-Wilks test) and variance homogeneity.

## 3. Results and Discussion

SOC was affected by land use up to a depth of 1m (Table 2). As land use produces changes in BD (Table 2), SOC content data has been corrected by BD. Mean SOC content up to 1m decreased as follows: forest (120.17 Mg ha$^{-1}$), pasture (94.57 Mg ha$^{-1}$) and cropped fields (81.82 and 76.49 Mg ha$^{-1}$, 6-9 years and >20 years cropping, respectively).There was a

significant reduction in SOC in the cropped plots compared to the forest in the first 20 cm and from 40 to 80 cm depth while pastures showed this decrease only in the surface layer. Between 34% and 48% of SOC was found in the first 20 cm, while in the forest it presented greater stratification (Jobbágy & Jackson, 2000). SOC vertical distribution tends to follow the distribution of the root system (Jobbágy & Jackson, 2000), the reason why pastures have a higher SOC is that roots are abundant down to a depth of 80-100 cm.

Land use had differential effects on SOC fractions (Figure 2). At both soil depths, coarse particulate carbon (2000 μm - 212μm, CPC) showed the greatest differences between treatments. Resistant organic carbon (<53μm, RC) was the main constituent of soil organic matter in all situations except in the forest, where CPC was the main fraction with 65% of total SOC in the superficial horizon and 55% in the 5-20 cm layer.

SOC content depended on the amount of carbon contributed by the vegetation that varied among vegetation types. Forest had

the greatest content, due to its higher net primary productivity, pastures represented the intermediate situation, and the lowest contribution corresponded to crops that in this region consist of one summer crop per year (Follet et al., 2009). Comparing




cropped and pristine soils in the Pampean region, Sainz Rozas et al. (2011) found that SOC reduction ranged between 36 and 53%, which placed our regional results in the middle of this range of variation. This loss of SOC can be explained by the lower contribution of crops, the greater mineralization and the greater susceptibility to erosion of these soils (Alvarez, 2001). The SOC fraction most affected by land use change was the most labile (CPC, 212µm -200µm), which represented 6% of

5 total SOC in cropped plots and 57% in pastures and forests. The higher CR content in the cropped plots (78% of total SOC), showed that there is a shift towards more humified fractions, which have a lower rate of nutrient mineralization, results that coincide with those obtained by Albanesi et al. (2003) and Galantini and Suñer (2008).

**Table 2: Soil organic carbon (SOC) and bulk density (BD) variation in depth associated with land use: forest, pasture, 6-9 years**
10 **and >20 years cropped soils. Different letters indicate significant differences between land use categories within each depth layer.**

| SOC (Mg ha$^{-1}$) | | | | | | | |
|---|---|---|---|---|---|---|---|
| | | | | | | *Cropped* | |
| *Depth (cm)* | *Forest* | | *Pasture* | | *6 - 9 years* | | *>20 years* | |
| 0 - 20 | 57.97 | a | 32.07 | b | 32.54 | b | 31.84 | b |
| 20 - 40 | 19.52 | a | 20.67 | a | 18.48 | a | 19.19 | a |
| 40 - 60 | 18.90 | a | 18.54 | a | 11.62 | b | 10.83 | b |
| 60 - 80 | 15.16 | a | 11.69 | b | 10.15 | bc | 7.89 | c |
| 80 - 100 | 8.62 | bc | 11.60 | a | 9.03 | b | 6.74 | c |
| 0 - 100 | 120.17 | a | 94.57 | b | 81.82 | bc | 76.49 | c |

| BD (Mg m$^{-3}$) | | | | | | | |
|---|---|---|---|---|---|---|---|
| | | | | | | *Cropped* | |
| *Depth (cm)* | *Forest* | | *Pasture* | | *6 - 9 years* | | *>20 years* | |
| 0 - 20 | 0.89 | c | 1.10 | b | 1.11 | b | 1.21 | a |
| 20 - 40 | 0.96 | c | 1.09 | b | 1.07 | b | 1.14 | a |
| 40 - 60 | 1.00 | d | 1.17 | a | 1.07 | c | 1.13 | b |
| 60 - 80 | 1.08 | a | 1.11 | a | 1.10 | a | 1.09 | a |
| 80 - 100 | 1.14 | a | 1.15 | a | 1.15 | a | 1.16 | a |

In soils with native forest, BD increased with depth, from 0.88 Mg m$^{-3}$ in the 0-20 cm layer to 1.14 Mg m$^{-3}$ at 80-100 cm depth. The cropped fields did not follow this trend. Their BD was highest in the surface layer (0-20 cm) and in depth (80-100 cm) and lowest at 20 to 80 cm. The pasture under 15 years of livestock production had similar values to the fields cropped

15 for 6-9 years. These higher surface BD values were a consequence of the decrease in SOC and machinery transit in cropped fields (Willhelm et al., 2004, Alvarez et al., 2012), and of the mechanical pressure exerted by livestock in the pastures





(Alvarez et al., 2012). Soils under more than 20 years of agriculture had a BD of 1.20 Mg m$^{-3}$ in the first 20 cm, 8% higher than soils under 6-9 years cropping or pasture (1.11 Mg m$^{-3}$) and 36% higher than the values of the forest soil (0.88 Mg m$^{-3}$). The highest soil aggregate MWD values were measured in forest (1.61 mm) and pasture (1.74 mm) soils, values that were not statistically different. Cropped plots showed an average MWD of 0.76 mm, half that of forest and pasture soils and

significantly (P<0.05) different to that of those two land-use categories. In all land use situations, fast wetting was the treatment that reduced MWD most, reducing the size of aggregates by 40% when compared with the treatment of less stress (slow wetting by capillarity). MWD, that characterizes structural stability, was directly related to CPC (r = 0.6, p = <0.01) and to SOC (r = 0.48, p <0.01). CPC loss of the first 20 cm largely explained the loss of MWD in cropped soils. Organic matter influences soil structure, but at the same time the formation of stabilized aggregates facilitates carbon sequestration

and provides physical protection to soil carbon (Onweremadu et al., 2007). However pastures, despite having a lower amount of coarse particulate carbon (212μm -200μm) than the forest, had the same MWD values. This could respond to the presence of a fine root network of the pasture grass (Gatton Panic) that improved aggregate resistance to stress.

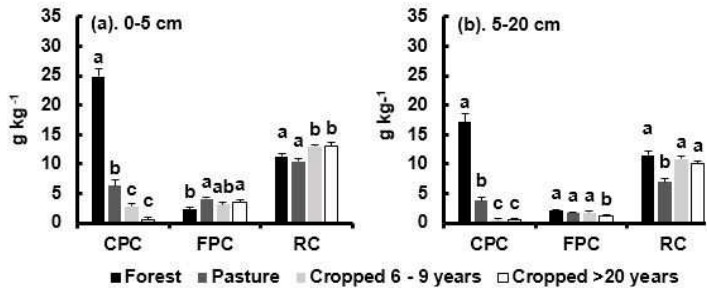

**Figure 2: Variation of coarse particulate carbon (2000 μm - 212μm, CPC), fine particulate carbon (212μm - 53μm, FPC) and resistant organic carbon (<53μm, RC) associated with land use. A: Depth 0-5 cm. B: Depth 5-20 cm. Different letters indicate significant differences between land use categories within each depth interval (P<0.05).**

Penetration resistance at 0-20 cm depth showed negative correlation with SWC (Figure 3A). PR of cropped fields was 1.1

MPa with 29% SWC, pastures had the same PR value at that depth but with lower SWC (22 g kg$^{-1}$). Forest showed higher PR values (1.5 MPa) as at the moment of sampling, SWC was lower (13 g kg$^{-1}$). At a greater depth (20-40 cm), no correlation was found between those two variables (Figure 3B). Fields under more than 20 years of cropping had an average PR of 3 MPa with high SWC values (27 g kg$^{-1}$). Soils with 6-9 years continuous cropping and forests had PR of 2.2 MPa, with a SWC of 25% and 14%, respectively. The pasture had the lowest PR (0.9 MPa) with a SWC of 21 g kg$^{-1}$. Below 20 cm

depth there was soil hardening in cropped plots; even with high SWC values (27 g kg$^{-1}$), their PR values werehigher than 2 MPa, which could be critical for root development. This would indicate soil compaction due to continuous machinery transit.




Lower than 1.50 MPa values at 0-20 cm depth could be attributed to the high organic matter content at that depth (Table 2 and Figure 3).

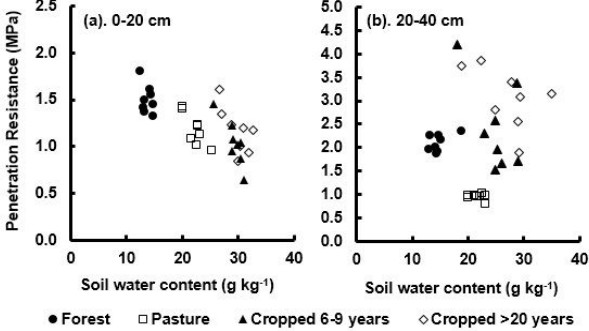

● Forest    □ Pasture    ▲ Cropped 6-9 years    ◇ Cropped >20 years

**Figure 3: Functional relationships between soil penetration resistance and gravimetric water content for different land uses at two**
**depths (A. 0-20 cm; B. 20-40 cm).**

**4. Conclusions**

In the study region, SOC content, its quality and distribution in the profile were sensitive to the change in land use. The conversion of Chaco forests to crop production was associated with SOC reductions up to one meter depth and with the decrease of the labile fraction, which occurred mainly in the first years after deforestation. The incorporation of pastures
proved to be a sustainable practice to mitigate C stock loss produced by cropping. It is of great interest to note that carbon losses are not restricted to the first few cm of the soil, as is generally shown in organic carbon maps or in greenhouse gas inventories. The development of models that would allow the estimation of SOC changes in depth would be useful to evaluate with greater precision the impact of land use change on carbon stocks. The change in land use also affected soil physical properties, such as compaction, loss of structural stability in the first 20 cm and hardening of the 20-40 cm depth
layer in fileds under no-till. Pastures, despite their lower SOC and CPC contents than the pristine soils, had a structural stability equal to that of the forest, showing that physical properties are not only correlated with the level of carbon in a soil, but also depend on the type of roots of the replacement vegetation and the stresses applied to the soil (i.e. machinery transit).

**5. Acknowledgments**

This work has been funded by UBACYT project No. 20020130100274BA. Farmers are thanked for their help in carrying out this work in their properties.





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
