# Peer review of "Effect of deforestation and subsequent land-use management on soil carbon stocks in the South American Chaco"

_SOIL, 2017_

## Short Comment (SC1) · 11 May 2018

The manuscript presents a good contribution to the knowledge of SOC stock distribution along the soil profile, as well as soil quality, according to land use changes and is of global interest, even if it was developed in the Chaco region of Argentina.

The scientific approach and the methods used are valid, well-known and reproducibles. The text is presented in a clear and well structured way and the Results and Discussion, as well as the Conclusions, are pertinent and expresses the work done and the explanations.

I would recommend the Manuscript to publication.

---

## Referee Comment (RC1) · P. G. Gottschalk (Referee) · 29 Jun 2018

The manuscript presents the results of changes in soil C stocks, bulk density, mean weight diameter and penetration resistance along a land use gradient in the Chaco of Argentina. Results are broken down to several soil layers down to 1 meter and different soil size fractions. Soil C stocks decrease from forest over pasture to long-term arable cropping with no difference between the two ages of the cropped sites.

General comments: The authors composed a compact manuscript with overall good quality. It is easy to read and mostly easy to comprehend. The concept as such is not new (tracking soil-C stock changes along chronosequences) but the compiled data

present another reference data set for land use changed induced soil C losses in a sensitive and crucial region of the world and is definitely suitable to publishing in SOIL. Objectives are clearly formulated. However, since the results are not unexpected under the given environmental changes I recommend to further improve the presentation of the results and the discussion and relate the results to similar studies (see suggested references below). The significance of the manuscript would benefit from the break-down of results to respective soil parameters (see detailed comment below). Although all sites are Haplustolls and Argiustolls and respective soil parameters are given in Table 1 and they are similar enough to classify them as one for the results' interpretation, it would be good to see the actual variability of the soil parameters (at sampled depth possibly) at the sites and how the soil C stocks and BD correlate with these. The soil parameters per sites could be given in the appendix and Table 1 then lists mean values and respective standard deviations. The discussion of MWD and penetration measurement results need to be improved in respect to an overall story line and the significance of the here presented results for land management and possible human interventions to improve soil quality. Overall, references to other studies where soil C-stock changes were analysed are missing, e.g. the studies of Johan Six, Karoline Denef or Balesdent should be included in respect of SOC distribution in different soil size fractions, e.g.: • Balesdent, J., et al. (1998). "The dynamics of carbon in particle-size fractions of soil in a forest-cultivation sequence." Plant and Soil 201: 49-57. • Six, J., et al. (2000). "Soil Structure and Organic Matter: I. Distribution of Aggregate-Size Classes and Aggregate-Associated Carbon." Soil Science Society of America Journal 64(2): 681-689. • Denef, K., et al. (2007). "Microaggregate-associated carbon as a diagnostic fraction for management-induced changes in soil organic carbon in two Oxisols." Soil Biology and Biochemistry 39(5): 1165-1172.

Specific comments: Lines 22-24 (abstract): This sentence suggests that the study investigated the effect of pasture as an intermediate phase during otherwise continuous cropping which is not true. Same formulation is used in the conclusions and should be adjusted. Please reference Mollisols, Haplustolls and Argiustolls as classified according to the USDA soil taxonomy or other but consistent. Page 3, line 14: of how many individual samples consisted one composite sample? Please specify. Page 4, line 5: Please elaborate shortly on the MWD-method, describe the method and how its specification makes it suitable for its designed purpose here. Please explain for what purpose the method is applied here, also for the penetration analysis. Please describe more carefully the sampling design and how "situation" (page 4, line 6, page 3, line 14) and "plot" (page 4, line 7) relates to each other. Page 4, lines 6-9: I do not understand why the sampling of penetration resistance and soil water content is not consistently sampled although the direct relation is explicitly mentioned. Please explain e.g. why the two samples of soil water content is sufficient in contrast to the penetration measurements every 5 cm. Page 4, line 14-16: Pasture C also decreased sig. in the layer 60-80 and increased sig. in layer 80-100. Please elaborate and discuss. The latter maybe due to the higher C inputs of grass roots in lower layers. Swap paragraphs 2 and 3 of the results and discussion section to keep the topics of SOC stocks versus SOC fractions apart. Page 4, line 21: Replace "treatment" with land use type or similar. Page 5, line 1-2: Add the soil depth for which the 36 and 53% soil C reductions is representative. For the discussion on the change of C in different soil size fractions check the papers of Balesdent at al., e.g. Balesdent, J., et al. (1998). "The dynamics of carbon in particle-size fractions of soil in a forest-cultivation sequence." Plant and Soil 201: 49-57.

Page 5, lines 12 – page 6, line 2: The discussion of BD values is a bit weak and not very conclusive. I suggest to at least adding the soil parameter description along the profile and discuss how soil texture could be related to the different BD values.

Page 6, lines 5-12: It is not clear what message the authors want to convey here and since the MWD-method has not been properly introduced it is difficult to follow a story line here. Page 6, line 19: Please add the R2-value and p-value of the negative correlation (possibly in the graphs of Figure 3). Page 6, line 19-26: Here, only the results a presented with no explanation or discussion. Please explain the significance

of the different penetration levels in respect to something, e.g. root growth, and relate the results to findings of other studies.

Technical corrections: Page 3, line 14: I suggest to replace "In each situation" with "at each site".

---

## Author Comment (AC1) · 19 Jul 2018

Response to Referee Comments:

General comment: Reviewer: Although all sites are Haplustolls and Argiustolls and respective soil parameters are given in Table 1 and they are similar enough to classify them as one for the results' interpretation, it would be good to see the actual variability of the soil parameters (at sampled depth possibly) at the sites and how the soil C stocks and BD correlate with these. The soil parameters per sites could be given in the appendix and Table 1 then lists mean values and respective standard deviations.

[Figure]

Authors: Haplustolls are dominant soils under forest and pasture management., while Hapludolls and Argiustolls are equally represented in cropped sites. Soil particle size distribution in all sites was only evaluated for the 0-20 cm layer (new Table). Soil BD was highly correlated with SOC (r= - 0.9; p>0.0001).

Mean values and standard errors of soil particle size distribution in the 0-20 cm layer for different land uses.

See New table as an attach file in Supplement.

Reviewer: Line 22-24 (Abstract) This sentence suggests that the study investigated the effect of pasture as an intermediate phase during otherwise continuous cropping which is not true. Same formulation is used in the conclusions and should be adjusted. Authors: We suggest (Abstract): "The permanent pasture of warm season grasses allowed to sustain higher C stocks than cropping systems and so could be considered a sustainable management practice". (Conclusions) "Permanent pasture of warm season grasses proved to be a sustainable practice to mitigate C stock loss compared with cropping systems".

Reviewer: Please reference Mollisols, Haplustolls and Argiustolls as classified according to the USDA Soil Taxonomy or other but consistent.

Authors: The Mollisol column should be deleted in Table 1 (since no other US Soil Taxonomy Soil Order was studied) and keeping the references to subgroup according to USDA Soil Taxonomy.

Reviewer: Page 3, line 14: of how many individual samples consisted one composite sample? Please specify. Authors: A composite sample built up from 4 subsamples.

Reviewer: Page 4, line 5: Please elaborate shortly on the MWD-method, describe the method and how its specification makes it suitable for its designed purpose here. Please explain for what purpose the method is applied here, also for the penetration analysis.

Authors: We suggest:

Aggregates of 3 to 5 mm in diameter were dried at 40 ° C for 24 hours and then subjected to three pre-treatments: fast wetting of air-dry aggregates with distilled water, wet agitation (previously treated with ethanol) and low wetting (capillarity with distilled water). After applying these pre-treatments, the distribution of the aggregates according to their size was determined using a series of sieves (0.05 mm, 0.1 mm, 0.2 mm, 0.5 mm, 1 mm and 2 mm). The aggregate mean weight diameter (MWD) for each pre-treatment was calculated as an index of the structural stability obtained as the algebraic sum of the percentage of the total mass of soil retained in each sieve, multiplied by the opening of the adjacent sieves. The MDW for each three pre-treatments was estimated and also an integrated value of MDW was calculated.

Erosion and compaction are the main soil degradation processes in the studied region. Soil MDW index was found to be conversely related to soil susceptibility to erosion. Soil penetration resistance allows characterizing soil compaction derived from machinery transit and animal trampling.

Reviewer: Please describe more carefully the sampling design and how "situation" (page 4, line 6, page 3, line 14) and "plot" (page 4, line 7) relates to each other.

Authors: We suggest changing "situation" or "plot" denominations by "site". We sampled 32 sites, eight by each management studied categories.

Reviewer: Page 4, lines 6-9: I do not understand why the sampling of penetration resistance and soil water content is not consistently sampled although the direct relation is explicitly mentioned. Please explain e.g. why the two samples of soil water content is sufficient in contrast to the penetration measurements every 5 cm.

Authors: Soil penetration resistance was determined up to 40 cm in each 5 cm soil player. Soil water content was only determined for the 0-20 cm and 20-40 cm layers. So, the relationship between soil resistance and water content was constructed by

integrating soil penetration resistance data for 0 to 20 cm and 20 to 40 cm layer depths.

Reviewer: Page 4, line 14-16: Pasture C also decreased sig. in the layer 60-80 and increased sig. in layer 80-100. Please elaborate and discuss. The latter maybe due to the higher C inputs of grass roots in lower layers.

Authors: Already discussed in page 4 lines 17-19 .

Reviewer: Swap paragraphs 2 and 3 of the results and discussion section to keep the topics of SOC stocks versus SOC fractions apart. Authors: Agree.

Reviewer: Page 4, line 21: Replace "treatment" with land use type or similar. Authors: Agree. We can change to land uses.

Reviewer: Page 5, line 1-2: Add the soil depth for which the 36 and 53% soil C reductions is representative. Authors: for 0-20 cm depth.

Reviewer: For the discussion on the change of C in different soil size fractions check the papers of Balesdent at al., e.g. Balesdent, J., et al. (1998). "The dynamics of carbon in particle-size fractions of soil in a forest-cultivation sequence." Plant and Soil 201: 49-57. Authors: Agree. Our results are similar to those of Balesdent et al. (1998), who found that the total C contents of soils decreased rapidly in the first seven years of cultivation and more slowly after and the decrease affected mostly the coarse CPC fractions. Balesdent, J., Besnard, E., Arrouays, D., Chenu, C. The dynamics of carbon in particle-size fractions of soil in a forest-cultivation sequence, Plant and Soil, 201, 49-57, 1998.

Reviewer: Page 5, lines 12 – page 6, line 2: The discussion of BD values is a bit weak and not very conclusive. I suggest to at least adding the soil parameter description along the profile and discuss how soil texture could be related to the different BD values.

Authors: As mentioned above, soil texture for all studied sites was only determined in the 0-20 cm layer. Soil SOC vs. BD relationship was also described before.

Reviewer: Page 6, lines 5-12: It is not clear what message the authors want to convey here and since the MWD-method has not been properly introduced it is difficult to follow a story line here.

Authors: We added a complete the description of the Le Bissonnais method for aggregate stability in the Materials and Methods section.

Reviewer: Page 6, line 19: Please add the R2-value and p-value of the negative correlation (possibly in the graphs of Figure 3). Page 6, line 19-26: Here, only the results a presented with no explanation or discussion. Please explain the significance of the different penetration levels in respect to something, e.g. root growth, and relate the results to findings of other studies.

Authors: Penetration resistance at 0-20 cm depth showed a negative correlation with SWC (r= -0.72, p<0.0001, Figure 3 A). At a greater depth (20-40 cm), no correlation was foynd between those variables (p= 0.32, Figure 3B). The practical significance of the values of PR were already explained and discussed in Page 6 line 25-26.

Reviewer:

Technical corrections: Page 3, line 14: I suggest to replace "In each situation" with "at each site". Authors: agree.

Thanks for you comments,

Please also note the supplement to this comment:
https://www.soil-discuss.net/soil-2017-34/soil-2017-34-AC1-supplement.pdf

**Supplement:**

Mean values and standard errors of soil particle size distribution in the 0-20 cm layer for different land uses.

|  | Clay (%) | Silt (%) | Sand (%) |
|---|---|---|---|
| Forest | 24±1.11 | 40±1.30 | 36±1.24 |
| Pasture | 24±1.58 | 39±2.62 | 37±2.81 |
| Cropped 6-9 years | 29±3.93 | 43±2.55 | 28±4.41 |
| Cropped >20 years | 30±2.24 | 44±4.77 | 26±5.21 |
| All land uses | 27±3.77 | 41±3.45 | 32.01±6.01 |

---

## Editor Comment (EC1) · V. Alcántara (Editor) · 20 Jul 2018

Dear authors, please note that the short comment posted to your manuscript (SC1) is also a comment from a selected referee. Please also address her comments in order to move forward with the review process. Many thanks and best regards, Viridiana Alcantara

---

## Author Comment (AC2) · 20 Jul 2018

First of all, thank you for your comment. The sub-humid Chaco region comprises a large area and there is scarce information about the impact of different land uses. Different land uses produced changes in SOC up to 1-m depth. It is important to know there magnitude in order to assess their impact on soil quality and climate change. On the other hand, a characterization of the physical quality of the soil was carried out. Monitoring soil physical quality is relevant in no-tillage systems, even more in silty soils. The silty texture soils have greater fragility against mechanical stress and low resilience.

---

## Author Comment (AC3) · 20 Jul 2018

Dear Viridiana, we have already answered both comments. We appreciate your collaboration. Thank you very much and best regards, Carina Alvarez

---

## Author Response (AR1)

Response to comments on "Effect of deforestation and subsequent land-use management on soil carbon stocks in the South American Chaco" by Natalia Andrea Osinaga et al. P. G. Gottschalk (Referee)

**Response to Referee Comments:**

*General comment:*
**Reviewer:** Although all sites are Haplustolls and Argiustolls and respective soil parameters are given in Table 1 and they are similar enough to classify them as one for the results' interpretation, it would be good to see the actual variability of the soil parameters (at sampled depth possibly) at the sites and how the soil C stocks and BD correlate with these. The soil parameters per sites could be given in the appendix and Table 1 then lists mean values and respective standard deviations.

**Authors:**
Haplustolls are dominant soils under forest and pasture management, while Hapludolls and Argiustolls are equally represented in cropped sites. Soil particle size distribution in all sites was only evaluated for the 0-20 cm layer (included in a new Table).
New Table. Table 2. Mean values and standard errors of soil particle size distribution in the 0-20 cm layer for different land uses. Page 4.

**Reviewer:** Line 22-24 (Abstract)
This sentence suggests that the study investigated the effect of pasture as an intermediate phase during otherwise continuous cropping which is not true. Same formulation is used in the *conclusions* and should be adjusted.
**Authors:**
We changed to (Abstract):
"The permanent pasture of warm season grasses allowed to sustain higher C stocks than cropping systems and so could be considered a sustainable management practice". Page 1, lines 23-24
We changed to (Conclusions):
"Permanent pasture of warm season grasses proved to be a sustainable practice to mitigate C stock loss compared with cropping systems". Page 8, lines 7-8.

**Reviewer:** Please reference Mollisols, Haplustolls and Argiustolls as classified according to the USDA Soil Taxonomy or other but consistent.

**Authors:** The Mollisol column was deleted in Table 1 (since no other US Soil Taxonomy Soil Order was studied) and keeping the references to subgroup according to USDA Soil Taxonomy. Page 4, Table 1.

**Reviewer:**
Page 3, line 14: of how many individual samples consisted one composite sample? Please specify.
**Authors:**
We added: A composite sample built up from 4 subsamples. Page 3, line 16.

**Reviewer:**
Page 4, line 5: Please elaborate shortly on the MWD-method, describe the method and how its specification makes it suitable for its designed purpose here. Please explain for what purpose the method is applied here, also for the penetration analysis.

**Authors:**
We added (Page 4, line 9-16):

Aggregates of 3 to 5 mm in diameter were dried at 40 ° C for 24 hours and then subjected to three pre-treatments: fast wetting of air-dry aggregates with distilled water, wet agitation (previously treated with ethanol) and low wetting (capillarity with distilled water). After applying these pre-treatments, the distribution of the aggregates according to their size was determined using a series of sieves (0.05 mm, 0.1 mm, 0.2 mm, 0.5 mm, 1 mm and 2 mm). The aggregate mean weight diameter (MWD) for each pre-treatment was calculated as an index of the structural stability obtained as the algebraic sum of the percentage of the total mass of soil retained in each sieve, multiplied by the opening of the adjacent sieves. The MDW for each three pre-treatments was estimated and also an integrated value of MDW was calculated.

We added (Page 5, line 2-5):

We measured MWD and penetration resistance as erosion and compaction are the main soil degradation processes in the studied region. Soil MDW index is conversely related to soil susceptibility to erosion and soil penetration resistance allows characterizing soil compaction derived from machinery transit and animal trampling.

**Reviewer:**
Please describe more carefully the sampling design and how "situation" (page 4, line 6, page 3, line 14) and "plot" (page 4, line 7) relates to each other.

**Authors:**
We changed "situation" or "plot" denominations by "site". We sampled 32 sites, eight by each management studied categories. Page 3, line16; page 4, lines 8 and 18; page 5, line 14; page 6, line 2; page 7, lines 2 and 23.

**Reviewer:**
Page 4, lines 6-9: I do not understand why the sampling of penetration resistance and soil water content is not consistently sampled although the direct relation is explicitly mentioned. Please explain e.g. why the two samples of soil water content is sufficient in contrast to the penetration measurements every 5 cm.

**Authors:**
Soil penetration resistance was determined up to 40 cm in each 5 cm soil player. Soil water content was only determined for the 0-20 cm and 20-40 cm layers. So, the relationship between soil resistance and water content was constructed by integrating

soil penetration resistance data for 0 to 20 cm and 20 to 40 cm layer depths. We clarified this in the text. Page 4 line 4 to page 5 line 2.

**Reviewer:**
Page 4, line 14-16: Pasture C also decreased sig. in the layer 60-80 and increased sig. in layer 80-100. Please elaborate and discuss. The latter maybe due to the higher C inputs of grass roots in lower layers.

**Authors:** Already discussed in page 5, lines 16-18 .

**Reviewer:**
Swap paragraphs 2 and 3 of the results and discussion section to keep the topics of SOC stocks versus SOC fractions apart.
**Authors:** Done.

**Reviewer:**
Page 4, line 21: Replace "treatment" with land use type or similar.
**Authors**: We can changed in all cases to land uses.

**Reviewer:**
Page 5, line 1-2: Add the soil depth for which the 36 and 53% soil C reductions
is representative.
**Authors:** for 0-20 cm depth. Added. Page 5, line 24.

**Reviewer:**
For the discussion on the change of C in different soil size fractions check the papers of Balesdent at al., e.g. Balesdent, J., et al. (1998). "The dynamics
of carbon in particle-size fractions of soil in a forest-cultivation sequence." Plant and Soil 201: 49-57.
**Authors:**
We added: "Our results are similar to those of Balesdent et al. (1998), who found that the total C contents of soils decreased rapidly in the first seven years of cultivation and more slowly after and the decrease affected mostly the coarse CPC fractions". Page 5 line 35 to page 6 line 2.
We also added the reference: Balesdent, J., Besnard, E., Arrouays, D., Chenu, C. The dynamics of carbon in particle-size fractions of soil in a forest-cultivation sequence, Plant and Soil, 201, 49-57, 1998. Page 9, line 11-12.

**Reviewer:**
Page 5, lines 12 – page 6, line 2: The discussion of BD values is a bit weak and not very conclusive. I suggest to at least adding the soil parameter description along the profile and discuss how soil texture could be related to the different BD values.

**Authors:**

As mentioned above, soil texture for all studied sites was only determined in the 0-20 cm layer (new Table 2).

We added: Finally, soil BD in 0-20 cm was highly correlated with SOC (r= - 0.9; p>0.0001) but not with soil particle size fractions. Page 6 line 16.

**Reviewer:**
Page 6, lines 5-12: It is not clear what message the authors want to convey here and since the MWD-method has not been properly introduced it is difficult to follow a story line here.

**Authors:**
We added a complete the description of the Le Bissonnais method for aggregate stability in the Materials and Methods section. Page 4, line 9-16.

**Reviewer:**
Page 6, line 19: Please add the R2-value and p-value of the negative correlation (possibly in the graphs of Figure 3).
Page 6, line 19-26: Here, only the results a presented with no explanation or discussion. Please explain the significance of the different penetration levels in respect to something, e.g. root growth, and relate the results to findings of other studies.

**Authors:**
We added: Penetration resistance at 0-20 cm depth showed a negative correlation with SWC (r=-0.72; p<0.0001; Figure 3A). Page 7, line 17.
At a greater depth (20-40 cm), no correlation was found between those two variables (p= 0.32; Figure 3B). Page 7 lines 19-20.
The practical significance of the values of PR were already explained and discussed in Page 7 line 23-25.

**Reviewer:**

Technical corrections: Page 3, line 14: I suggest to replace "In each situation" with "at each site".
**Authors:** done. Page 3, line 16.

[revised manuscript text omitted]

---

## Author Response (AR2)

Response to comments on "Effect of deforestation and subsequent land-use management on soil carbon stocks in the South American Chaco" by Natalia Andrea Osinaga et al. Viridiana Alcantara /Topical Editor. Thanks for your corrections and dedication!!!!!!

5 Editor: Change to 1970s. Authors: Done. Page 3, Line 13.
Editor: Change to no-till. Authors: Done. Page 3, Line 13.
Editor: Add (c). Authors: Done. Page 3, Line 14.
Editor: Sensitive to what? Decomposition?
Authors: Changed to: The coarse particle fraction (2000 um 212 um) at 0.

Authors: Changed to: The coarse particle fraction (2000 µm -212 µm) at 0-5 cm and 5-20 cm depth layers was the most

10 sensitive organic carbon fraction to land use change. Page 3, Line 18.

Editor: Better responsive. Authors: Done. Page 3, Line 21.

Editor: organic. Authors: Done. Page 3, Line 22.

Editor: Change to 1. Authors: Done. Page 3, Line 23.

Editor: Land use system in terms of C preservation. Authors: Done. Page 3, Line 24-25.

15 Editor: Change to C. Authors: Done. Page 3, Line 25, 26, 27.

Editor: Land use. Authors: Done. Page 3, Line 28.

Editor replacement through what? Suggest complete de sentence as follows:

...a process that involves replacement of natural ecosystems, such as forests, by agricultural land (cropping or grassland systems) as world food demand increases (Volante et al., 2012).

20 Authors: Done. Page 3, Line 33.

Editor: Thus, Authors: Done. Page 4, Line 2.

Editor: forest-cleared o deforested. Author: Done. Page 4, Line 4.

Editor: leads to. Authors: done. Page 4, Line 7.

Editor: consider changing to warming. Authors: Done. Page 4, Line 9.

25 Editor: American. Authors: done. Page 4, Line 10.

Editor: Could you also added here the enhanced soil C stocks in no-till systems compared to conventional till is mostly limited to the first cm of soil.

Authors Done. Page 4, Line 17.

Editor: soil compaction caused by animal transit. Author: Done. Page 4, Line 24.

- 30 Editor: More precise: stocks. Authors: Changed to stocks. Page 4, Line 27.
  - Editor: Argentina, Authors: done. Page 4, Line 2.
  - Editor: . A new sentence. Authors: done. Page 5, Line 15.
  - Editor: . and begin a new sentence. Authors: Done. Page 5, Line 18.

Editor: in the abstract you mention you compare using equivalent mass, please explain here how it was calculated. Authors: included. The citation was also included in the Reference section. Page 5, Line 1-2. Editor: Significance of differences was testing. Authors: done. Page 7, Line 4. Editor: delete this citation at this position. Authors: done. Page 7, Line 14.

- 5 Editor: input. Authors: done. Page 7, Line 17.
  Editor: Please indicate the most important crops for this region.
  Authors: done. The most important annual crops in the region are soybean, maize and cotton. Page 7, Line 19-20.
  Editor: lower C input due to harvesting. Authors: done. Page 7, Line 22-23.
  Editor: different. Authors: done. Page 7, Line 24.
  10 Editor: land uses. Authors: done Page 7, Line 26.
- Editor: Iand uses. Authors: done Page 7, Line 26.
  Editor: Please specify 0-5 cm and 5-20 cm. Authors: done. Page 7, Line 27.
  Editor: cropping. Authors: done. Page 8, Line 9.
  Editor: land uses (delete situations) Authors: done. Page 8, Line 15.
  Editor: As a contrast. Authors: done. Page 9, Line 3.
- Editor: depth, Authors: done. Page 9, Line 18.
  Editor: SOC stock. Authors done. Page 10, Line 6.
  Editor: Would you add to your conclusions to recommend to strengthen efforts of reducing deforestation in the South American Chaco?

Authors: Agree. We added: We recommend to strengthen efforts to minimize or stop deforestation in the South American

20 Chaco, as remaining native forests provide numerous essential ecosystem services such as carbon sequestration, climate and flood regulation, and preservation of biodiversity. Page 10, 11 to 13.

**Effect of deforestation and subsequent land-use management on soil carbon stocks in the South American Chaco**

Natalia Andrea Osinaga1; Carina Rosa Álvarez2, Miguel Angel Taboada1,2,3

[revised manuscript text omitted]

|                     | Area | Horizon  | Clay Silt             |                       | Sand                  | II  |
|---------------------|------|----------|-----------------------|-----------------------|-----------------------|-----|
|                     | (%)  | HOLIZOII | (g kg -1 ) | (g kg -1 ) | (g kg -1 ) | рп  |
| Typic
Haplustoll | 80   | А        | 20                    | 43                    | 37                    | 6.6 |
|                     |      | AC       | 35                    | 47                    | 18                    | 6.7 |
| Typic
Argiustoll | 20   | А        | 25                    | 47                    | 28                    | 6.8 |
|                     |      | Bt       | 41                    | 40                    | 19                    | 6.6 |

Table 1: Main characteristics of the soils of the study region.

[revised manuscript text omitted]